# *Polyscias filicifolia* (*Araliaceae*) Hairy Roots with Antigenotoxic and Anti-Photogenotoxic Activity

**DOI:** 10.3390/molecules27010186

**Published:** 2021-12-29

**Authors:** Anita Śliwińska, Ramona Figat, Anna Zgadzaj, Beata Wileńska, Aleksandra Misicka, Grzegorz Nałęcz-Jawecki, Agnieszka Pietrosiuk, Katarzyna Sykłowska-Baranek

**Affiliations:** 1Department of Pharmaceutical Biology and Medicinal Plant Biotechnology, Faculty of Pharmacy, Medical University of Warsaw, 1 Banacha St., 02-097 Warsaw, Poland; anita.sliwinska@wum.edu.pl (A.Ś.); agnieszka.pietrosiuk@wum.edu.pl (A.P.); katarzyna.syklowska-baranek@wum.edu.pl (K.S.-B.); 2Department of Environmental Health Science, Faculty of Pharmacy, Medical University of Warsaw, Banacha 1, 02-097 Warsaw, Poland; azgadzaj@wum.edu.pl (A.Z.); gnalecz@wum.edu.pl (G.N.-J.); 3Faculty of Chemistry, University of Warsaw, 1 Pasteura St., 02-093 Warsaw, Poland; bwilenska@chem.uw.edu.pl (B.W.); misicka@chem.uw.edu.pl (A.M.); 4Biological and Chemical Research Centre, 101 Żwirki i Wigury St., 02-097 Warsaw, Poland

**Keywords:** elicitation, caffeic acid derivatives, ferulic acid derivatives, cytotoxic activity, HPLC-PDA-ESI-MS analysis

## Abstract

Hairy root cultures are considered as a valuable source of bioactive phytoconstituents with expanding applicability for their production. In the present study, hairy root cultures of *Polyscias filicifolia* (*Araliaceae*), a traditional Southeast Asian medicinal plant, were established. The transformation with *Agrobacterium rhizogenes* ATCC 15834 allowed to obtain 15 root lines. The K-1 line, demonstrating the highest growth capabilities, was subjected to further investigations. To enhance the biosynthetic potential of hairy roots, methyl jasmonate elicitation approach was applied (MeJA; at different doses and exposure time), with subsequent transfer of elicited roots to control medium. This strategy resulted in chlorogenic acid production up to 1.59 mg/g dry weight. HPLC-PDA-ESI-MS analysis demonstrated variation in extracts composition and allowed to identify different caffeic and ferulic acid derivatives. Next, cytotoxic, antigenotoxic, and anti-photogenotoxic properties of hairy roots extracts were determined. None of the tested extracts were cytotoxic. In addition, they demonstrated significant antigenotoxic activity with the highest protective potential; up to 52% and 49% of inhibition of induction ratio (IR) induced by the 2-aminoanthracene was revealed for extracts derived from hairy roots elicited for 3 days with 50 µM MeJA and roots elicited for 7 days with 100 µM MeJA and then transferred for 30 days to control medium, respectively. These same extracts exhibited the highest anti-photogenotoxic potential, up to 36% of inhibition of chloropromazine-induced genotoxicity.

## 1. Introduction

Plant biotechnological methods offer the possibility for production of a wide array of various bioactive compounds for pharmaceutical, as well as food and cosmetic industries in a manner independent of environmental conditions, at the same time ensuring protection of biodiversity of natural resources [1]. Among various systems of cultivated in vitro cell and organs, hairy root cultures are recognized as a reasonable system for efficient production of complex specific metabolites, as they are capable of producing the same profile of metabolites as a native plant, or even de novo synthesizing biomolecules [2,3,4,5]. Many methods are used in plant biotechnology to increase the efficiency of the production of specific metabolites in cells, tissues, and organs of plants grown in in vitro cultures. One of these methods is elicitation. In response to elicitor stress, the production of plant-specific metabolites increases. Specific metabolites resulting from elicitation are mainly responsible for protecting plants against pathogens, attack by herbivores and insects, as well as for protection in a stressful situation [6].

*Polyscias filicifolia* (C. Moore ex E. Fourn.) L.H. Bailey is a member of the *Polyscias* genus that comprises 159 species, most of which are shrubs or trees naturally growing in South Asia [7]. The plants of genus *Polyscias* are used in traditional medicine to treat various ailments, such as obesity, malaria, and mental-illness among others. They are reported to possess anti-inflammatory, antibacterial, and adaptogenic properties. Up to now, 97 compounds belonging to various phytochemical groups have been isolated form species of *Polyscias* genus, including sterols, phenolic compounds, essential oils, and saponins, mainly possessing oleanolic acid as a aglycone [8]. In addition, one representative of the genus i.e., *P. filicifolia* was introduced into in vitro cultures and the callus and shoot cultures were developed to examine their morphogenetic potential [9,10,11]. Simultaneously, the investigation of chemical composition and biological properties of callus and shoots in vitro was performed. The major phytochemical constituents were represented by phenolic compounds and demonstrated antioxidant, antibacterial, antimutagenic, and anti-photogentoxic properties [10,12,13,14]. Furthermore, extracts of shoots elicited with 200 µM methyl jasmonate (MeJA) improved HaCaT (immortal keratinocyte from adult human skin) cell viability [10].

So far, the development of hairy root cultures was reported only in genus *Panax (P. ginseng* [15], *P. quinquefolium* [16,17,18,19]) and *Aralia* (*A. elata* [20]) representatives of the *Araliaceae* family.

This study reports, for the first time, the method of establishing *Polyscias filicifolia* hairy root cultures and investigation of changes in their phytochemical profile affected by elicitation. Furthermore, cytotoxic, antigenotoxic, and anti-photogenotoxic properties of plant extracts were estimated.

## 2. Results

### 2.1. Hairy Root Cultures

Hairy root development was observed only after transformation of shoots regenerated from apical meristems and plantlets derived from somatic embryos using *Agrobacterium rhizogenes* ATCC 15834 strain. No roots were observed when *A. rhizogenes* LBA 9402 strain was used (Figure 1; Table 1).

The addition of acetosyringone (Acs) to the MS medium had a beneficial effect on the transformation efficiency. In the case of shoots incubated on MS medium with Acs, the percentage of explants on which roots were formed was the highest and amounted to 65%. These roots emerged directly (without the callus phase) 4 weeks after the inoculation. The average number of roots developed per shoot explant was 4.8. In comparison, the transformation efficiency of explants cultivated on the MS medium without the addition of Acs was found to be 20%, with the average number of emerging roots per shoot explant being 2.9 (Table 1).

The transformation efficiency of plantlets derived from somatic embryos was lower than that of shoots. On the MS medium supplemented with Acs, root formation was found at the site of infection in 25% of infected plants; the average number of emerging roots per explant was 2.5. On the MS medium without the addition of Acs, only 5% of the plants demonstrated root formation, and the average number of roots per explant was 1.7 (Table 1).

Finally, 15 root lines were obtained, and each root that sprouted was cut off and cultivated separately as a single line. Among all obtained root lines, only three were selected for further research: K-1, K-3, and K-5 lines (Figure 2). Observation of the growth and morphology of these three root lines through three passages led to the final selection of the one K-1 line that was characterized by an abundant hairy zone without any callus formation. The roots of the other 12 lines did not sustain growth due to necrosis. In addition, the formation of an abundant callus was observed on the roots of the K-3 and K-5 lines. The transformation process was confirmed by PCR analysis (Appendix A).

#### Optimalisation of Culture Conditions

The most hairy root growth-promoting medium was selected on the basis of root growth index, their anatomical traits, and color. Among the tested medium modifications, the DCR I medium with L-glutamine 500 mg/L and 30 g/L sucrose proved to be the most growth promoting (Figure 3 and Figure 4). During cultivation, the medium hairy roots of K-1 line demonstrated a growth index up to 480.8% ± 23.63. Roots were white, had numerous side branches, and formed a dense clump. These roots did not change color throughout the duration of the passage (Figure 3).

After various modifications of WPM medium, the root growth was significantly lower compared to the DCR I medium. The modification of WPM medium by the addition of L-glutamine 500 mg /L resulted in an improvement of root growth capacities in comparison to the WPM medium without this amino acid (Figure 3). Contrary to the DCR I medium, with all the modifications of the WPM media, regardless of the presence of L-glutamine and at high sucrose concentration (40 g/L), the root darkening started after 3 weeks of culture and they became dark brown after 6 weeks.

### 2.2. Phytochemical Analysis

Among four qualitatively investigated phenolic acids by the HPLC-UV-Vis analysis, only the presence of chlorogenic acid (CGA) in extracts prepared from hairy roots cultivated in control conditions, as well as in elicited cultures, was revealed (Figure 5).

In control untreated cultures, the CGA biosynthesis was significantly lower than in elicited cultures. In addition, the increase in CGA accumulation was correlated with the time of culture (Figure 5). The significantly (*p* < 0.05) highest content of CGA (1.59 ± 0.06 mg/g DW) was determined and elicited for 1 day with methyl jasmonate (MeJA) 100 µM culture followed by subsequent cultivation in control DCR I medium. It was over 12% higher concentration than that detected in hairy roots treated for 7 days with MeJA 100 µM and then cultivated for 30 days in control DCR I medium. The content of CGA in hairy roots elicited with MeJA 50 or 100 µM for 3 or 7 days, respectively, was also higher than in control unelicited culture, although no statistically significant differences in CGA concentration were observed.

The HPLC-PDA-ESI-MS analysis of hairy root methanol extracts derived from unelicited and elicited cultures revealed variation in the qualitative profile of the samples (Figure 6; Table 2). The major constituents of investigated extracts were caffeic and ferulic acid derivatives (Table 3). For all standard samples, mass spectra and fragmentation mass spectra were acquired for identification and confirmation of compounds presented in the methanolic extracts. In that case, a HRMS experiment was also used for confirmation of molecular formula. For the unknown compounds a HRMS experiment was the only one method, which was applied for prediction of the most likely molecular formula. In all HRMS experiments, a difference between theoretical and measurement *m*/*z* value was below 5 ppm.

### 2.3. Cytotoxicity of Hairy Root Extracts

None of tested samples decreased BALB/c 3T3 cells’ viability after 24 h treatment under 70% (Figure 7 and Figure 8). Moreover, none of tested samples induced changes in the cells morphology in comparison to the untreated control. Therefore, all of the tested samples were classified as non-cytotoxic in vitro in the whole range of tested concentrations. Due to the lack of cytotoxicity, the IC50 values that represent the concentrations that decreased the number of living cells to 50% in comparison to the untreated control, were not calculated. However, there were observed slight but statistically significant (Student’s *t*-test, *p* < 0.05) differences in the cells’ viability compared with the negative control for the highest concentrations of several samples (the bars with asterisks on the Figure 7). This effect was not observed for any solutions prepared from samples collected after 50 days of the in vitro plant culture. Furthermore, on the basis of obtained results, there were no clearly visible differences among samples prepared with various solvents (methanol, 50% methanol or deionized water). The results obtained for the solvent control confirmed the lack of its influence on the cells viability. Moreover, the results obtained for the positive control fulfilled the required conditions (IC50 = 115 µg/mL) [21].

The results obtain for the presented samples and their dilution series (concentrations from 50 to 0.78 µg/mL; results unshown) did not differ from the viability of cells observed in the untreated control (Student’s *t*-test, *p* < 0.05).

### 2.4. Antigenotoxic and Anti-Photogenotoxic Activities

The antigenotoxic and anti-photogenotoxic potentials of methanolic extracts prepared from hairy roots cultivated under various conditions were investigated. The dry residue of extracts were dissolved in methanol, 50% methanol, and deionized water. Three types of solvents were used to test whether they would affect the activity due to different degrees of dissolution of the compounds. Furthermore, methanol is toxic to the bacteria used in the *umu* test. The use of water as a solvent allowed higher concentrations of extracts to be tested.

In the preliminary *umu*-test, the photogenotoxicity and genotoxicity of the tested extracts was evaluated. None of the tested concentrations exhibited a genotoxic effect (IR < 1.5).

The results of the *umu*-test performed with UV radiation against CPZ-induced photogenotoxicity are presented in Table 4 and Figure 9. Root extracts were added to the mutagen at concentrations of 100–800 μg/mL (extracts dissolved in water) or 12.5–100 μg/mL (dissolved in methanol or 50% methanol). Lower concentrations of the solutions were prepared in the case where the solvent was methanol, because of the toxicity of methanol against bacteria. In the highest tested concentration of water solutions (Table 4), all of the extracts were active against CPZ. Extracts F and H exhibited the highest anti-photogenotoxic potential, up to 36% of inhibition of CPZ-induced genotoxicity. In the lowest tested concentrations of water solutions (100 μg/mL), all of the extracts were active as well, and the most active was extract I. When extracts were dissolved in methanol (Table 4), in all tested concentration extracts, G, H, and I exhibited anti-photogenotoxic potential. Extract F was not active only in the concentration of 25 μg/mL. In the case of extract C, the increase of genotoxicity was observed in the concentration of 50 μg/mL. The other tested extracts have no significant activity in the methanolic solutions. When extracts were dissolved in 50% methanol (Table 4), only four of them exhibited activity (F, G, H and I), and extract F had the strongest protective effect against CPZ-induced photogenotoxicity. A comparison of all the extracts tested at the same concentration (100 μg/mL) using different solvents is shown in Figure 9.

The results of the *umu*-test performed with metabolic activation against 2AA-induced genotoxicity are presented in Table 5 and Figure 10. In the highest tested concentration of water solutions (Table 5), only extract A did not exhibit antigenotoxic activity. The highest protective effect was observed for extract F and I, up to 52% of inhibition of IR induced by the 2AA. In the lowest tested concentration of water solutions (100 μg/mL), four extracts exhibited activity (F, G, H, and I), and the most active was extract F. Extract F, when dissolved in methanol (Table 5), was active in all tested concentrations. The methanolic solutions of extracts G and D exhibited a protective effect in the concentration of 100 μg/mL. The increase of 2AA-induced genotoxic effect was observed in the treatment with D in the concentration of 12.5 μg/mL. When extracts were dissolved in 50% methanol (Table 5), only extract I was active in all tested concentrations. Extracts F and G were active in the lowest tested concentration. Extracts B and C exhibited no antigenotoxic potential. Extracts A and D, when dissolved in 50% methanol, exhibited a protective effect in the two tested concentrations. Figure 10 presents a comparison of the activity of different extracts (at the same concentration: 100 μg/mL) dissolved in three different solvents. Extracts F, G, H, and I exhibited higher activity when dissolved in deionized water. Extract D was only active when dissolved in pure methanol or 50% methanol.

## 3. Discussion

Hairy root cultures, due to their genetic and biosynthetic stability and substantial biomass proliferation capabilities are considered as an abundant system for the production of bioactive compounds [4,22,23]. The optimization of cultural conditions allowed us to achieve the productivity of specific metabolites at levels comparable or even higher than native plants; moreover, hairy roots were reported to biosynthesize phytochemicals that are accumulated only in aerial parts or even absent in the parent plant [24]. This study describes, for the first time, the establishment of *Polyscias filicifolia* hairy root cultures. Up to now, the hairy root cultures were reported only for several members of the *Araliaceae* family. In the present study, the development of hairy roots was observed on micropropagated and somatic embryo-derived shoots transformed with *A. rhizogenes* ATCC 15834 (Table 1). The application of acetosyringone, which was reported to increase the efficiency of T-DNA transfer into plant cell [2], exerted a beneficial effect on transformation frequency in the current study (Table 1). Jeong et al. [15] reported transformation process of filed-grown *Panax ginseng* roots by *A. rhizogenes* KCTC 2744, and Kochan et al. [16] transformed filed-grown seedlings of *P. quinquefolium* with *A. rhizogenes* ATCC 15834. In vitro-derived plant material was used for the transformation of *P. quinquefolium,* where the best response was obtained by the transformation of epicotyls with *A. rhizogenes* LBA 9402 [18] and *Aralia elata* with the highest transformation rate observed on root and petiole explants infected with *A. rhizogenes* ATCC 15834 [20]. In the mentioned reports for the cultivation of hairy roots, mostly modifications of B5 [25] or MS [26] media were used. Whereas, in the present study the highest biomass increase was noted during hairy root cultivation in DCR I [27] (Figure 4), with an increase to 400 mg/L MgSO_4_ and supplemented with L-glutamine 500 mg/L. DCR I medium was originally developed for the cultivation of *Taxus* × *media* hairy roots [28].

The previous investigations on hairy root cultures of *Araliaceae* species focused mostly on the production of ginsenosides. In the current study, the accumulation of several phenolic acids was examined, and among the others CGA demonstrated the highest concentration (1.59 ± 0.06 mg DW) in hairy roots elicited on 1 day with 100 µM MeJA 100 and then transferred to control medium for 30-day cultivation (extract H). However, the CGA content was over 3-fold lower than reported previously in *P. filicifolia* shoots elicited with 50 µM salicylic acid [10]. Although subsequent HPLC-PDA-ESI-MS analysis showed that CGA is present in the form of its derivative, this is not surprising as CGA was demonstrated to easily transform or degrade to its derivatives [29].

The approach applied in the present study, relying on the elicitation and subsequent transfer of elicited roots for further cultivation to control medium, proved to be superior to elicitation exclusively (Figure 5). A similar approach was used by Sivanandhan et al. [30] to enhance withanolides production in *Withania somnifera* hairy root cultures, resulting in a remarkable increase in their production, although the authors did not compare the effectiveness of this approach to the elicitation itself. Nevertheless, this approach allows to obtain a significant increase in hairy root productivity by biosynthesis stimulation without retardation in biomass proliferation, which is often a reported effect of elicitor treatment [31].

Furthermore, in the present study, methanolic extracts resulted from hairy roots cultures were subjected to HPLC-PDA-ESI-MS analysis. The composition of examined extracts varied in relation to the applied treatments. Oleanolic acid was detected only in control cultures up to 2 weeks. While CGA, 5-*O*-feruoylo-quinic, and 3,5-dicaffeoyl-quinic acids were observed since the third week of culture in control and elicited cultures (Table 2). Other phenolic acids i.e., caffeic, *p*-coumaric, ferulic, and sinapic were not detected. Further extract profiling revealed the presence of caffeic and ferulic acid derivatives, which was also reported for methanolic extracts of *P. filicifolia* shoots [10,14]. These shoot extracts demonstrated considerable antimutagenic, anti-photogenotoxic, and skin cell-protective activity [10,12,13,14].

In the present study, three types of solvents were used to test whether they would affect the activity due to different degrees of dissolution of the compounds. The influence on cell viability of hairy root methanolic extracts was assessed, and no cytotoxic effect was demonstrated (Figure 6 and Figure 7). This finding corresponds with the previous results reported for *P. filicifolia* shoot [10] as well callus extracts [12]. Next, photogenotoxicity and genotoxicity of the tested extracts was evaluated by the preliminary *umu*-test. The results indicate that none of the concentrations tested exhibited a genotoxic effect. Furthermore, in experiments with CPZ, the highest activity of extracts was observed when they were dissolved in water. It could be concluded that observed biological activity was affected by the presence of polar phenolics, which were determined in investigated extracts (Table 4). In addition, almost all extracts, except A, exerted antigenotoxic activity with the highest protective potential determined for extracts F and I, up to 52% of inhibition of IR induced by the 2AA (Figure 9; Table 5). The observed biological activities could be attributed to the presence of caffeic and ferulic acid derivatives, rather than CGA isomer presence; the highest concentration was noted in H extract as caffeic and ferulic acid derivatives, which were reported to be more active [32,33]. It was shown that starting from the 30 day of culture, the enhancement in biosynthetic capacities of hairy roots was observed; in all extracts, 3,5-dicaffeoylquinic and derivative of 5-*O*-feruoyloquinic acid were determined. Nevertheless, further phytochemical and biological investigations should be carried out to reveal the full application potential of *P. filicifolia* hairy root extracts.

## 4. Materials and Methods

### 4.1. Hairy Root Cultures

#### 4.1.1. Plant Material

The plant material subjected for transformation consisted of: (i) 7-day-old leaves, petioles, separated from a 2-year-old *P. filicifolia* plant; (ii) embryogenic callus (1 g) obtained from leaves cultivated on Murashige and Skoog solid medium (MS, [26]) supplemented with 2 mg/L 2,4-dichlorophenoxyacetic acid (2,4-D) and 0.01 mg/L kinetin (KIN); (iii) plantlets derived from somatic embryos, cultivated on MS/2 medium without growth regulators with 15 g /L sucrose [9]; (iv) shoots regenerated from apical meristems of plants growing on Linsmaier and Skoog solid medium (LS, [34]) supplemented with 2 mg/L BAP and 0.5 mg/L KIN [10].

A voucher specimen of *P. filicifolia* was deposited at the Department of Pharmaceutical Biology and Medicinal Plants Biotechnology, Faculty of Pharmacy, Medical University of Warsaw, Warsaw, Poland (accession No. FW21/026/1999).

#### 4.1.2. Establishment of Hairy Root Cultures

Genetic transformation was performed with the use of two strains: *Agrobacterium rhizogenes*: ATCC 15834 and LBA 9402. Bacteria were grown in the dark on solid YMB medium [35] at 26 °C. Prior to transformation, bacteria were incubated for 48 h in liquid medium YMB with or without the addition of 0.2 mg/L acetosyringone (Acs). Acs, dissolved in a drop of 96% ethanol, was added to the autoclaved medium by filtering through a sterile filter (Ø 20 µm pores).

The explants were contaminated by pricking them with a sterile needle dipped in the bacterial suspension. Leaves were infected along the nerve, and petioles at their central part. Plantlets derived from somatic embryos and micropropagated in vitro shoots were pricked several above the second node. After infection, the explants were placed on solid hormone-free MS medium (25 mL of medium in 100 mL Erlenmeyer flasks) or on MS medium supplemented with 0.2 mg/L Acs. After 4 weeks, the percentage of explants on which root growth was observed was calculated, and the number of roots formed per one explant was determined.

After 4 weeks of cultivation on solid MS medium, roots approx. 10 mm long were cut off explants and transferred individually to 100 mL Erlenmeyer flasks containing 25 mL of MS liquid medium supplemented with Biotaxine 500 mg/L. The flasks were placed on the rotary shaker (100 rpm INNOVA 2300 platform shaker, New Brunswick Scientific, Enfield, CT, USA) and cultivated in the dark. After two passages with the antibiotic, the concentration of Biotaxine added to the medium was reduced to 250 mg/L. After four passages, a test was performed for the presence of bacteria in the cultured roots. For this purpose, their fragments were placed on solid YMB and incubated for 48 h at 24 °C in the dark. The lack of bacterial growth indicated the sterility of the obtained transformed roots. The sterile roots were transferred to 250 mL Erlenmeyer flasks containing 50 mL of hormone-free MS liquid medium and routinely subcultured every 4–6 weeks. As a result of the performed transformation, 15 clones of hairy roots were obtained. Finally, only one hairy root line K-1 was selected for further investigation.

#### 4.1.3. DNA Extraction and PCR Analysis

In *P. filicifolia* roots obtained after transformation of *Agrobacterium rhizogenes* with ATCC 15834 strain, the presence of *rol*B, *rol*C, and *vir*G genes was analyzed using the PCR technique. The primer sequences used to carry out the PCR reaction are shown in Table 6. The PCR reaction was performed using the Phire^®^ Plant Direct PCR Kit (Finnzymes, Thermofisher Scientific, Waltham, MA, USA) according to the manufacturer’s instruction. The leaves of non-transformed *P. filicifolia* donor plant were used as a control. About 3 mm of putative hairy root or control leaves were cut off and suspended in 20 μL of dilution buffer, and 0.5 μL of resulted solution was taken for PCR reaction. The PCR reaction was carried out in a volume of 20 μL, under the conditions according to the protocol. PCR analysis conditions were as follows: one cycle of initial denaturation at 98 °C for 5 min; the next 40 cycles at 98 °C for 5 s, 62 °C for 5 min, and 72 °C for 20 s; and then a final extension at 72 °C for 1 min. The Thermal Cycler Vario (Applied Biosystems, Waltham, MA, USA) for amplification was used. The products were electrophoretically separated on a 1.0% agarose gel (PRONA, Burgos, Spain) with the 1 × TBE buffer using SimplySafe (EURx, Gdańsk, Poland) and UV light visualised (Gel-X Gel Analysis Systems, STI, Poznań, Poland). GeneRuler^TM^ 1 kb Plus DNA ladder (Thermo Scientific, Waltham, MA, USA) molecular weight marker was used to estimate the size of amplification products.

#### 4.1.4. Cultivation of Hairy Roots and Elicitation

In order to select the best growth-promoting medium for the cultivation K-1 hairy root for line, the following hormone-free media modifications were tested: (i) DCR (Gupta and Durzan [27]) with 500 mg/L L-glutamine; (ii) DCR medium with L-glutamine 500 mg/L and increased to 400 mg/L MgSO_4_ × 7 H_2_O [28] and 40 g/L sucrose (DCR I); (iii) Woody Plant Medium (WPM; Lloyd and McCown [37]) with 30 g/L sucrose; (iv) WPM medium supplemented with 500 mg/L L-glutamine and 30 g/L sucrose (WPM I); (v) WPM medium with 40 g/L sucrose (WPM II); and (vi) WPM medium supplemented with 500 mg/L L-glutamine and 40 g/L sucrose (WPM III).

The growth of the root biomass was estimated based on fresh (FW) and dry weight (DW) (Appendix A). The 28-day-old hairy roots of K-1 line were taken out of the flasks and gently pressed on a filter paper, drained, and weighed to record the fresh biomass. Next, the frozen roots were lyophilised (lyophilizer Christ ALPHA1-4 LSC, Osterode am Harz, Germany) until constant weight, and the dry weight (DW) was recorded. The root growth index was calculated based on fresh biomass of roots using the following formula: (final weight-initial weight/initial weight) × 100% [38].

Hairy root culture of K-1 line were elicited after 3 or 7 days with 50 or 100 µM methyl jasmonate (MeJA CAS no. 39924-52-2; Sigma-Aldrich, Poznań, Poland). In addition, hairy roots elicited with 100 µM MeJA for 1 or 7 days were transferred to liquid DCR I medium (40 mL in 250 mL Erlenmeyer flask).

### 4.2. Antigenotoxicity and Antiphotogenotoxicity Assesment Chemicals

2-aminoanthracene (2AA) and chloropromazine (CPZ) were purchased from Sigma-Aldrich (Poznań, Poland). 2AA were dissolved in DMSO. CPZ was dissolved in deionized water. D-glucose 6-phosphate disodium salt hydrate (G-6-P) (CAS no. 3671-99-6), disodium salt hydrate (CAS no. 3671-99-6), and 2-nitrophenyl β-d-galactopyranoside (ONPG) (CAS no. 369-07-3), which was the β-galactosidase enzyme, were purchased from Sigma Aldrich. Nicotinamide adenine dinucleotide phosphate (NADP) (CAS no. 24292-60-2) was purchased from MP Biomedicals (Irvine, CA, USA). DMSO (CAS no. 67-68-5) was purchased from Avantor Performance Materials (Gliwice, Poland). Methanol (CAS no. 67-56-1) was purchased from Merck (Warsaw, Poland).

#### Antigenotoxicity and Anti-Photogenotoxicity Assessment by the *umu*-Test

The antigenotoxic potential was evaluated against the genotoxic action of 2AA in the presence of metabolic activation. Metabolic activation was performed with an S9 liver fraction of a male Sprague Dawley rat treated with Aroclor 1254 (single dose of 500 mg/kg body weight) in soya oil 5 days before the isolation according to the method described by Maron and Ames [39].

Firstly, the genotoxic potentials of 2AA (10 μg/mL) was measured by the *umu*-test using *Salmonella typhimurium* strain TA1535/pSK1002 (Deutsche Sammlung von Mikroorganismen und Zellkulturen GmbH DSMZ, Braunschweig, Germany). The *umu*-test was carried out in the micro-plate variant according to the ISO guideline [40]. β-galactosidase activity of the tested extracts was presented as an induction ratio (IR) calculated as the β-galactosidase activity of the tested extract relative to the negative control. The spectrophotometric measurement was performed at λ = 420 nm to indicate the intensity of the enzymatic reaction, and the optical density at λ = 600 nm indicated the bacteria growth. The measurements were performed with an Asys UVM340 Hightech microplate spectrophotometer. The 1.5-fold and greater increase in the β-galactosidase activity resulted in an IR value of 1.5 or greater and indicated the genotoxicity of the sample. Simultaneously, the genotoxicity of extracts (800 μg/mL or 100 μg/mL) was investigated. Next, the inhibition of 2AA genotoxicity by tested extracts was investigated. Extracts were added to the mutagen at concentrations of 100–800 μg/mL (extracts dissolved in water) or 12.5–100 μg/mL (dissolved in methanol or 50% methanol). Lower concentrations of the solutions were prepared in the case where the solvent was methanol, since methanol is toxic to bacteria.

The rate of antigenotoxicity (%) of the tested sample was calculated as the inhibition of IR induced by 2AA at the particular concentration according to the following Equation:Antigenotoxicity% = (1 − IR_genotoxin+sample_/IR_genotoxin_)100%(1)

To assess the anti-photogenotoxic effect of tested extracts, the protocol developed by Skrzypczak et al. [41] was applied. Chlorpromazine (CPZ) was used as an agent-inducing genotoxic response under UVA irradiation by forming an unstable promazyl radical, which is able to bind to DNA [42]. CPZ was tested at the concentration of 5 μg/mL by the modified *umu*-test after UVA irradiation was carried out with the lamp emitting UVA 365 nm (0.231 mW/cm^2^). The rate of anti-photogenotoxicity was calculated using the same formula as the rate of antigenotoxicity.

### 4.3. Determination of Cytotoxicity Using Neutral Red Uptake Test

The neutral red uptake cytotoxicity test was performed on the basis of modified ISO 10993 guideline Annex A [21] with BALB/c 3T3 clone A31 mammalian cell line (mouse embryonic fibroblasts from American Type Culture Collection). The quantitative estimation of viable cells in tested culture was based on their neutral red uptake in comparison to the results obtained for untreated cells. Dead cells have no ability to accumulate the dye in their lysosomes.

The BALB/c 3T3 cells were seeded in 96-well microplates (10^4^ cells/100 µL) in DMEM (Gibco, Thermofisher Scientific, Waltham, MA, USA) culture medium (supplemented with 10% of calf bovine serum, 100 IU/mL penicillin and 0.1 mg/mL streptomycin) and incubated for 24 h (5% CO_2_, 37 °C, >90% humidity). At the end of the incubation, each well was examined under a microscope to ensure that cells form a half-confluent monolayer. After that, culture medium was replaced by the tested solutions, previously sterilized by filtration and 100 times diluted in the fresh culture medium with reduced serum concentration (5%). The highest tested concentration of each sample in the culture medium was 100 µg/mL. Cells were treated with eight dilutions of each solution in a 2-fold dilution series for 24 h (three data points for each variant). Subsequently treatment medium was removed. Cells were washed with PBS and treated with the neutral red medium for 2 h. Then the medium was discarded, and cells were washed with PBS and treated with desorbing fixative (ethanol and acetic acid water solution). The amount of neutral red accumulated by cells was evaluated colorimetrically at 540 nm. As negative and positive controls, fresh culture medium and sodium lauryl sulphate (SLS) were used, respectively. The solvent control was fresh DMEM with 1% of methanol (the concentration of this alcohol in DMEM varies among three variants of tested solutions from 1% to 0%). The percentage of viable cells in each well was calculated by comparing its OD_540_ result with the mean result obtained for untreated cells (incubated in the same conditions with fresh culture medium). When the BALB/c 3T3 cells’ viability was not decreased under 70% by the sample in the selected concentration, it was considered as non-cytotoxic in this assay.

### 4.4. Phytochemical Analysis Chemicals

Methanol (CAS no. 67-56-1), acetonitrile (CAS no. 75-05-8), and acetic acid (CAS no. 64-19-7) of HPLC Grade were purchased in Merck (Germany). Deionized water was obtained from Simplicity Millipore water purification system (Merck, Germany). While standard substances of chlorogenic (CAS no. 327-97-9), caffeic (CAS no. 331-39-5), ferulic (CAS no. 1135-24-6), 3,5-dicaffeoyl-quinic (CAS 2450-53-5), oleanolic acid (CAS 508-02-1), sinapic (CAS 530-59-6), and *p*-coumaric (CAS) acids were purchased in Sigma-Aldrich (Poznań, Poland); 5-*O*-feruoylouinic acid (CAS) 1135-24-6) was purchased in LCG Standards (Lomianki, Poland). Methanol (CAS no. 67-56-1) for plant material extraction was purchased from Avantor Performance Materials (Gliwice, Poland).

#### 4.4.1. Sample Preparation and HPLC-UV-Vis Analysis

Hairy roots cultivated in DCR I medium were subjected to phytochemical analysis. Hairy roots derived from the following culture variants were examined: (A) control roots collected after 10 days of culture; (B) control roots collected after 20 days of culture; (C) control roots collected after 30 days of culture; (D) control roots collected after 40 days of culture; (E) control roots collected after 50 days of culture; (F) hairy roots elicited for 3 days with 50 µM MeJA; (G) hairy roots elicited for 7 days with 100 µM MeJA; (H) hairy roots elicited for 1 day with 100 µM MeJA 100 and then transferred to control medium for 30 day cultivation; and (I) hairy roots elicited for 7 days with 100 µM MeJA 100 and then transferred to control medium for 30-day cultivation.

The extraction of 100 mg of lyophilized hairy roots was performed according to the protocol describe by Śliwińska et al. [10]. In brief, plant material was extracted with 100% methanol (3 × 1 mL) for 15 min in an ultrasonic bath (Sonorex Bandelin, Berlin, Germany). Extracts were collected, evaporated to dryness, and stored at −20 °C. HPLC analysis was performed using DIONEX HPLC system (Sunnyvale, Sunnyvale, CA, USA) according to the method described by Śliwińska et al. [10]. In brief, C18 EC 250/4.6 Nucleosil 120–127 mm reversed phase column (Macherey-Nagel, Düren, Germany) was used and gradient elution (A solvent 0.04 M orthophosphoric acid; B solvent acetonitrile): 0 min, B10%; 5 min, B45%; and 15 min, B55% with flow rate 1 mL/min, was applied.

Quantitative determination of chlorogenic acid (CGA), caffeic acid (CA), ferulic acid (FA), and *p*-coumaric acid (PCA) was performed.

#### 4.4.2. LCMS-IT-TOF Analysis

A Shimadzu Prominence high-performance liquid chromatograph (HPLC) was used coupled with a LCMS-IT-TOF mass spectrometer (Shimadzu, Kyoto, Japan), equipped with an ion trap (IT), a time-of-flight (TOF) detector, and an electrospray ionization (ESI) source. Mass spectra were recorded in the positive and negative ion modes using LCMS solution software (Shimadzu, Kyoto, Japan).

Conditions for HPLC separation and detection of extracts were as follows: column Kinetex C_18_, 2.6 µm, 2.1 × 100 mm (Phenomenex, Torrance, CA, USA); injection volume: 3 µL; oven column temperature: 40 °C; flow rate: 0.2 mL/min; analysis duration: 75 min; and PDA detection at wavelengths λ = 200–800 nm. The mobile phase consisted of (A) water with the addition of 0.2% CH_3_COOH and (B) methanol. The following gradient was applied: 0–10 min 5% B, 10–30 min 5→50% B, 30–35 min 50→50% B, 35–55 min 50→95% B, 55–60 min 95% B, 60–62 min 95→5% B, and equilibrium time—13 min in 5% B.

Conditions for the mass spectrometer were as follows: polarity positive and negative; mass range *m*/*z* 100–1000 Da in both modes; ion accumulation time: 10 ms in MS1 experiments and 25ms in MS2 experiments; interface temperature: 220 °C; heat block temperature: 220 °C; nebulizing gas flow: 1.5 L/min; drying gas pressure: 100 kPa, IS: +4.5 kV for positive mode and −3.5 kV for negative mode; and collision energy in MS2 experiments: 25–35%.

The calibration mixture was used to calibrate the TOF detector of the LCMS-IT-TOF mass spectrometer for high resolution mass spectrometry experiments (HRMS). The sample was prepared by dissolving in the methanol and spinning on a vortex, and the supernatant was transferred to an HPLC injection vial.

### 4.5. Statistical Analysis

The statistical analyses were made on the results of at least three independent biological replicates performed with completely fresh bacteria cultures (all measurements were performed in triplicate or more independent experiments). All data were analyzed using the STATISTICA 13.1 PL software package (StatSoft, Kraków, Poland). For data with normal distribution (tested with Shapiro–Wilk test), Student’s *t*-test was used. Data without normal distribution were analyzed with the nonparametric U Mann–Whitney method. The results were considered to be statistically significantly different at a probability level of *p* < 0.05.

## 5. Conclusions

The effectiveness of root formation on *P. filicifolia* explants infected with *Agrobacterium rhizogenes* depended on: the bacterial strain used, the type of explant, and the presence of acetosyringone (Acs) in the medium on which the infected explants were incubated. The approach of elicitation with subsequent hairy root transfer to control medium proved to be the most promoting in relation to the intensification of phenolic acid biosynthesis. Methanolic extracts derived from *P. filicifolia* hairy roots exhibited no cytotoxic effect while their antigenotoxic and anti-photogenotoxic properties were demonstrated. The most active proved to be dissolved in water extracts derived from hairy roots cultivated in elicited variants of cultures. This could be attributed to the higher level of phenolic compounds such as chlorogenic acid, and dicaffeoylquinic and feruoylcaffeoylquinic acid derivatives, which were more numerous and abundant than in control roots. The results obtained in the current study indicate the future direction of further investigations of *P. flilicifolia* towards its photoprotective capabilities.

## Figures and Tables

**Figure 1 molecules-27-00186-f001:**
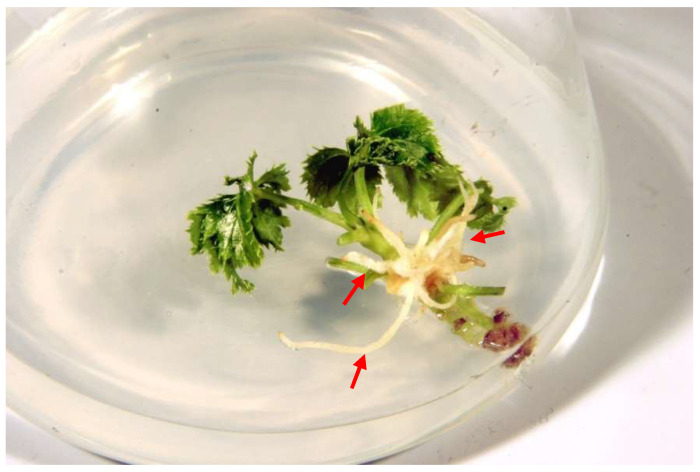
Roots emerging from infected *Agrobacterium rhizogenes* ATCC 15834 shoots after 4 weeks of culture. Shoots were cultivated on hormone-free MS solid medium supplemented with acetosyringone.

**Figure 2 molecules-27-00186-f002:**
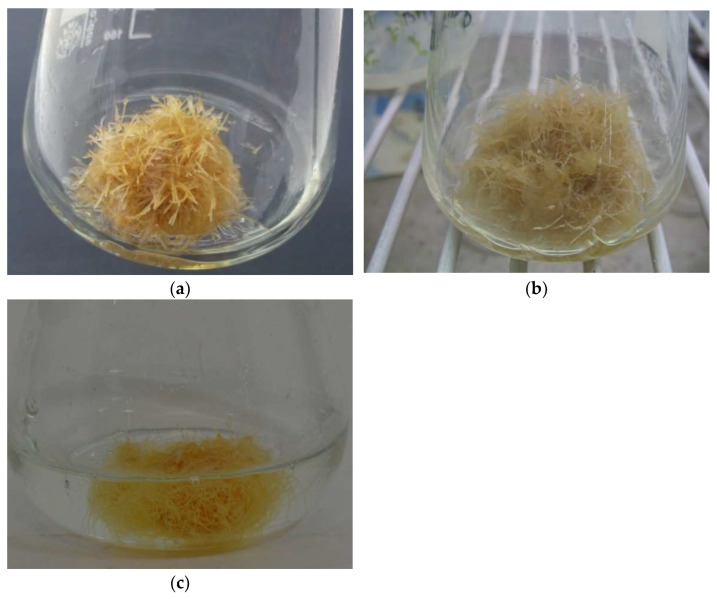
Three selected hairy root lines of *Polyscias filicifolia* obtained as a result of transformation performed with *Agrobacterium rhizogenes* ATCC 15834 strain. Hairy roots were cultivated in liquid hormone-free MS medium: (**a**) K-1 line; (**b**) K-3 line; (**c**) K-5 line.

**Figure 3 molecules-27-00186-f003:**
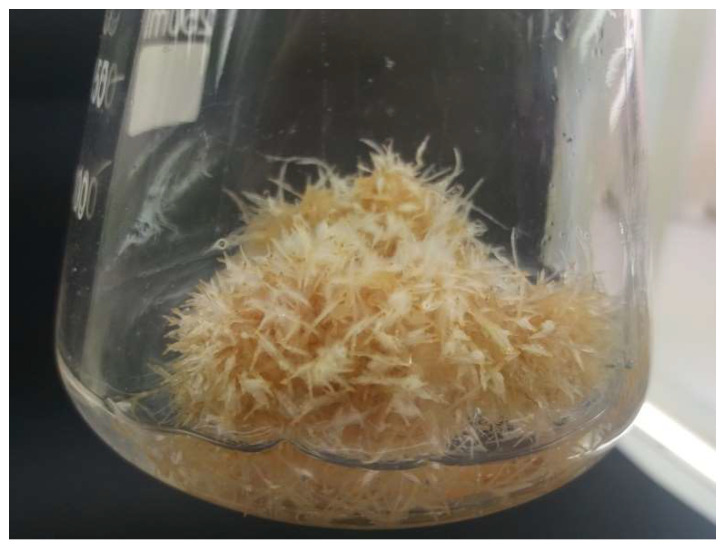
Hairy roots of K-1 line growing in liquid DCR I medium supplemented with L-glutamine 500 mg/L, 30 g/L sucrose.

**Figure 4 molecules-27-00186-f004:**
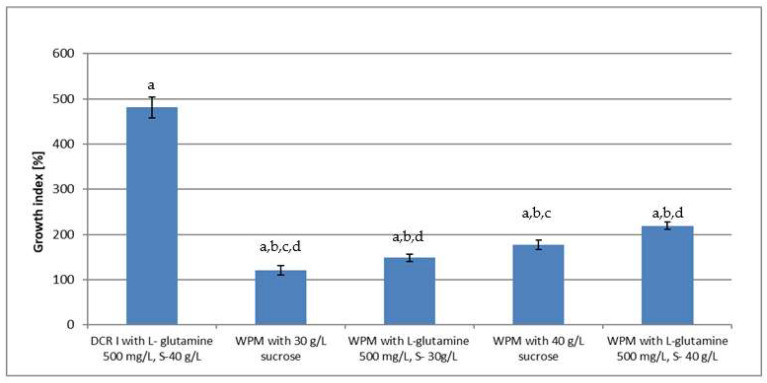
The comparison of K-1 hairy root line fresh biomass growth expressed as growth index [%] is calculated as follows: (final weight-initial weight/initial weight) × 100%. The same letter indicates a statistically significant difference (*p* < 0.05).

**Figure 5 molecules-27-00186-f005:**
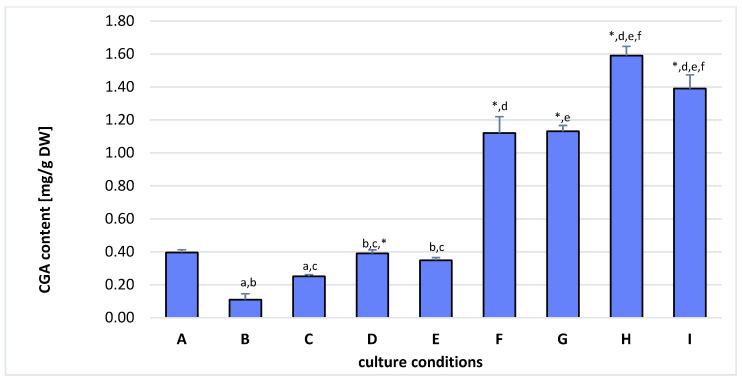
Chlorogenic acid (CGA) content [mg/g DW] determined in hairy roots of K-1 line cultivated under various conditions: (**A**) control roots collected after 10 days of culture; (**B**) control roots collected after 20 days of culture; (**C**) control roots collected after 30 days of culture; (**D**) control roots collected after 40 days of culture; (**E**) control roots collected after 50 days of culture; (**F**) hairy roots elicited for 3 days with 50 µM MeJA; (**G**) hairy roots elicited for 7 days with 100 µM MeJA; (**H**) hairy roots elicited for 1 day with 100 µM MeJA 100 and then transferred to control medium for 30 day cultivation; (**I**) hairy roots elicited for 7 days with 100 µM MeJA 100 and then transferred to control medium for 30 day cultivation. Means denoted with the same letter or asterisks are statistically significant (*p* < 0.05).

**Figure 6 molecules-27-00186-f006:**
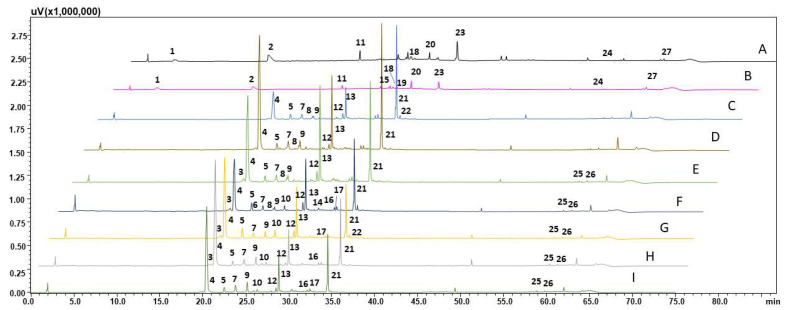
HPLC-PDA-ESI-MS chromatograms of methanolic extract derived from hairy roots cultivated under various culture conditions: (**A**) control roots collected after 10 days of culture; (**B**) control roots collected after 20 days of culture; (**C**) control roots collected after 30 days of culture; (**D**) control roots collected after 40 days of culture; (**E**) control roots collected after 50 days of culture; (**F**) hairy roots elicited for 3 days with 50 µM MeJA; (**G**) hairy roots elicited for 7 days with 100 µM MeJA; (**H**) hairy roots elicited for 1 day with 100 µM MeJA 100 and then transferred to control medium for 30 day cultivation; (**I**) hairy roots elicited for 7 days with 100 µM MeJA 100 and then transferred to control medium for 30 day cultivation.

**Figure 7 molecules-27-00186-f007:**
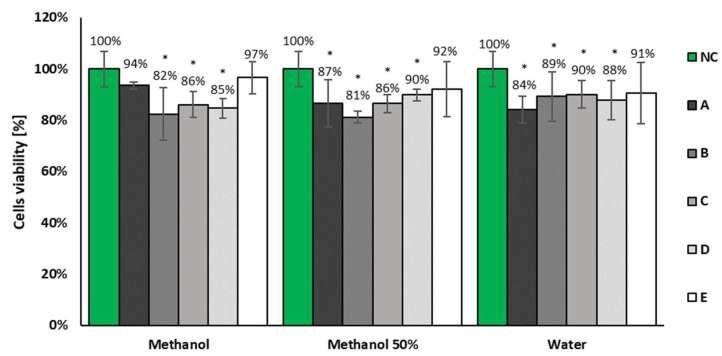
The results of the neutral red uptake assay with BALB/c 3T3 cells after 24 h of exposition for the highest concentrations of tested samples (100 µg/mL): plant culture extracts dissolved in methanol, 50% methanol or deionized water; collected every 10 days of the in vitro plant culture. NC—the viability of cells treated only with fresh DMEM medium; (**A**, **B**, **C**, **D**, and **E**) means: extracts from hairy roots collected from untreated culture after 10, 20, 30, 40, and 50 days. * The asterisks highlights the statistically significant (Student’s *t*-test, *p* < 0.05) differences in the cells’ viability compared with the negative control. The results obtained for the dilution series of all tested samples (concentrations from 50 to 0.78 µg/mL; results not shown) did not differ from the viability of cells observed in the untreated control.

**Figure 8 molecules-27-00186-f008:**
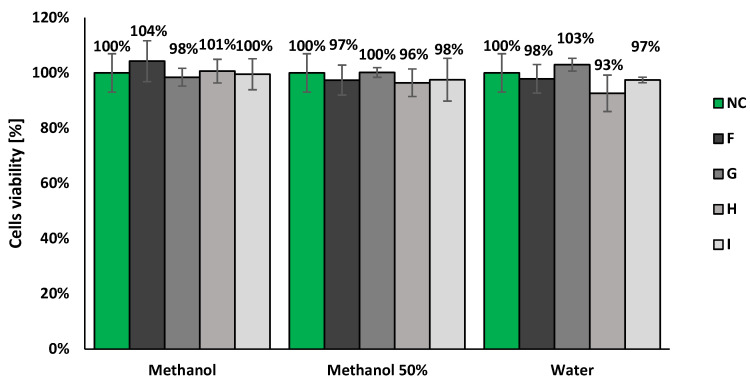
The results of the neutral red uptake assay with BALB/c 3T3 cells after 24 h of exposition for the highest concentrations of tested samples (100 µg/mL): plant culture extracts dissolved in methanol, 50% methanol, or deionized water; obtained from the in vitro plant culture treated with various elicitors. NC—the viability of cells treated only with fresh DMEM medium; (**F**) hairy roots elicited for 3 days with 50 µM MeJA; (**G**) hairy roots elicited for 7 days with 100 µM MeJA; (**H**) hairy roots elicited for 1 day with 100 µM MeJA 100 and then transferred to control medium for 30 day cultivation; (**I**) hairy roots elicited for 7 days with 100 µM MeJA 100 and then transferred to control medium for 30 day cultivation.

**Figure 9 molecules-27-00186-f009:**
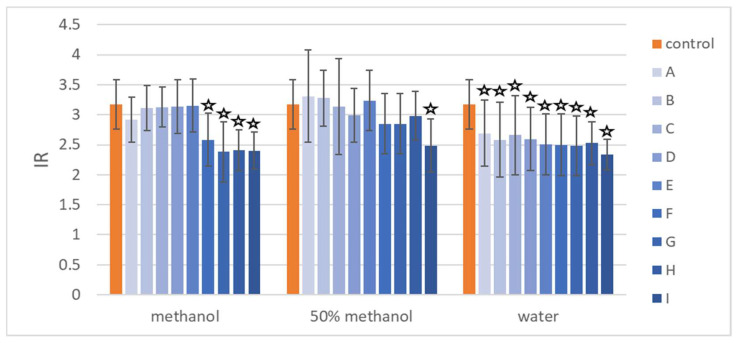
The results of the evaluation of anti-photogenotoxic activity based on IR values for the control—photogenotoxin chlorpromazine (CPZ 5 µg/mL)—and the mixture of CPZ with extracts (100 µg/mL) dissolved in methanol, 50% methanol, or deionized water. The asterisks highlights the statistically significant (*p* < 0.05) differences in the cells’ viability compared with the control. The extracts from the following treatments were investigated: (**A**) control roots collected after 10 days of culture; (**B**) control roots collected after 20 days of culture; (**C**) control roots collected after 30 days of culture; (**D**) control roots collected after 40 days of culture; (**E**) control roots collected after 50 days of culture; (**F**) hairy roots elicited for 3 days with 50 µM MeJA; (**G**) hairy roots elicited for 7 days with 100 µM MeJA; (**H**) hairy roots elicited for 1 day with 100 µM MeJA 100 and then transferred to control medium for 30 day cultivation; (**I**) hairy roots elicited for 7 days with 100 µM MeJA 100 and then transferred to control medium for 30 day cultivation.

**Figure 10 molecules-27-00186-f010:**
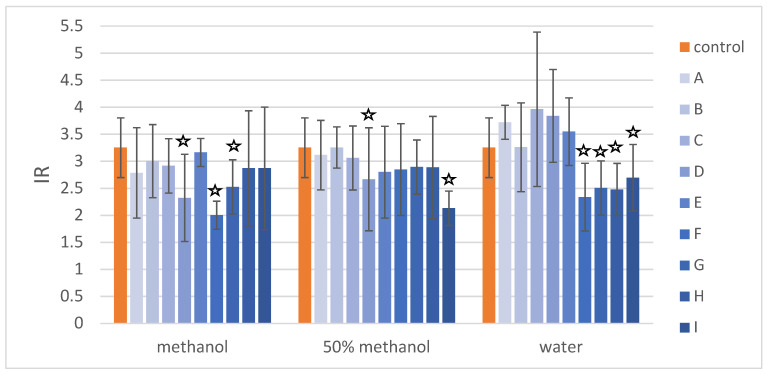
The results of the evaluation of antigenotoxic activity based on IR values for the control—genotoxin 2-aminoanthracene (2AA 10 µg/mL)—and the mixture of 2AA with extracts (100 µg/mL) dissolved in methanol, 50% methanol, or deionized water. The asterisks highlights the statistically significant (*p* < 0.05) differences in the cells viability compared with the control. The extracts resulted from the following treatments were investigated: (**A**) control roots collected after 10 days of culture; (**B**) control roots collected after 20 days of culture; (**C**) control roots collected after 30 days of culture; (**D**) control roots collected after 40 days of culture; (**E**) control roots collected after 50 days of culture; (**F**) hairy roots elicited for 3 days with 50 µM MeJA; (**G**) hairy roots elicited for 7 days with 100 µM MeJA; (**H**) hairy roots elicited for 1 day with 100 µM MeJA 100 and then transferred to control medium for 30 day cultivation; (**I**) hairy roots elicited for 7 days with 100 µM MeJA 100 and then transferred to control medium for 30-day cultivation.

**Table 1 molecules-27-00186-t001:** Efficiency of hairy root formation on various explants infected with *Agrobacterium rhizogenes* ATCC 15834 strain and incubated on MS medium with or without acetosyringone (Acs).

Medium	Type of Explant	% of Explants with Emerging Roots	Average Number of Roots Growing per One Explant
MS	7-day-old leaves from donor plant	0%	0
MS + Acs		0%	0
MS	7-day-petioles old from donor plant	0%	0
MS + Acs		0%	0
MS	Embryogenic callus	0%	0
MS + Acs		0%	0
MS	Plantlets derived from somatic embryos cultivated in vitro	5%	1.7
MS + Acs		25%	2.5
MS	Shoots derived from apical meristems cultivated in vitro	20%	2.9
MS + Acs		65%	4.8

**Table 2 molecules-27-00186-t002:** HPLC-PDA-ESI-MS analysis of hairy root extracts based on standard compounds used in present study.

Tested Extract	Caffeic Acid	Chlorogenic Acid Isomer	*p*-Coumaric Acid	5-*O*-Feruoylo-quinic Acid Isomer	Sinapic Acid	Ferulic Acid	3,5-Dicaffeoyl-quinic Acid	Oleanolic Acid
A	−	−	−	−	−	−	−	+
B	−	−	−	−	−	−	−	+
C	−	+	−	+	−	−	+	−
D	−	+	−	+	−	−	+	−
E	−	+	−	+	−	−	+	−
F	−	+	−	+	−	−	+	−
G	−	+	−	+	−	−	+	−
H	−	+	−	+	−	−	+	−
I	−	+	−	+	−	−	+	−

Extracts of: (A) control roots collected after 10 days of culture; (B) control roots collected after 20 days of culture; (C) control roots collected after 30 days of culture; (D) control roots collected after 40 days of culture; (E) control roots collected after 50 days of culture; (F) hairy roots elicited for 3 days with 50 µM MeJA; (G) hairy roots elicited for 7 days with 100 µM MeJA; (H) hairy roots elicited for 1 day with 100 µM MeJA 100 and then transferred to control medium for 30 day cultivation; (I) hairy roots elicited for 7 days with 100 µM MeJA 100 and then transferred to control medium for 30-day cultivation.

**Table 3 molecules-27-00186-t003:** HPLC-PDA-ESI-MS data on detected compounds in hairy root extracts derived from various cultures.

Peak No.	Tr	[M − H]^−^	Molecular Formula	Compound	Condition
1	4.94	253	C9H18O8	galactosylglycerol	A, B
2	15.94	378	C15H25NO10	unidentified	A, B
3	19.93	325 341	C15H18O8 C15H18O9	*p*-cumaric acid glucoside caffeic acid 3-glucoside	E, F E, F, G, H
4	20.43	353, 707 (dimer)	C16H18O9	chlorogenic acid (isomer)	C, D, E, F, G, H, I
5	22.41	353	C16H18O9	1/3/5-*O*-caffeoylquinic acid (chlorogenic acid)	C, D, E, F, G, H, I
6	22.82	387	C18H29O9	tuberonic acid glucoside	F
7	23.73	337	C16H18O8	5-*p*-coumaroylquinic acid	C, D, E, F, G, H, I
8	24.9	551	C26H32O13	ladroside	C, D, E, F
9	25.08	367	C17H20O9	5-*O*-feruloylquinic acid (isomer)	C, D, E, F, G, H, I
10	26.28	473	C21H30O12	4-allyl-2-(beta-d-glucopyranosyloxy)-phenyl beta-d-glucopyranoside	F, G, H, I
11	26.59	367	C17H20O9	3-*O*-feruoylquinic acid	A, B
12	28.49	440 515 463	C21H31NO9 C25H24O12 C21H36O12	unidentified isochlorogenic acid A unidentified	C, D, E C, D, E, F, G, H, I G, H, I
13	28.85	515	C25H24O12	3,5-dicaffeoyl-quinic acid	C, D, E, F, G, H, I
14	30.16	479	C25H28N4O6	unidentified	F
15	31.97	529	C26H26O12	3,4-*O*-(*E*)-caffeoylferuoylquinic acid	B
16	32.06	457 475 505	C23H38O9 C30H36O5 C23H38O12	unidentified lespeflorin B unidentified	F, H, I F, H, I H, I
17	32.35	373	C18H30O8	unidentified	F, G, I
18	32.51	407 529	C19H24N2O8 C26H26O12	unidentified 3,4-*O*-feruoylcaffeoylquinic acid	A, B A, B
19	32.98	459	C34H36O	unidentified	B
20	34.67	314 577	C16H29NO5 C30H26O12	unidentified catechin derivative (procyanidin B1–B6, B8)	A, B A, B
21	34.74	577	C30H26O12	catechin derivative	C, D, F, G, H, I
22	35.15	535	C28H40O10	unidentified	C, G
23	37.92	591	C31H28O12	8,8′-methylenebiscatechin	A, B
24	56.37	455	C30H48O3	oleanolic acid	A, B
25	58.96	605	C35H58O8	curculigosaponin B	E, F, G, H, I
26	59.89	531	C32H52O6	rubianol D	E, F, G, H, I
27	61.59	545	C33H54O6	unidentified	A, B

**Table 4 molecules-27-00186-t004:** Evaluation of anti-photogenotoxic activity based on IR values for the control—photogenotoxin chlorpromazine (CPZ 5 µg/mL)—and the mixture of CPZ with extracts dissolved in methanol, 50% methanol, or deionized water.

Tested Extract						Concentration of Tested Extracts [µg/mL] IR Values (Mean ± SD)//%Antigenotoxicity									
	0	12.5		25		50		100		200		400		800	
**Control**	**3.17 ± 0.41**														
**A**	methanol	2.97 ± 0.55	-	3.36 ± 0.26	-	3.36 ± 0.37	-	2.92 ± 0.37	-						
**B**		2.87 ± 0.52	-	3.17 ± 0.26	-	3.46 ± 0.77	-	3.11 ± 0.37	-						
**C**		2.90 ± 0.39	-	3.18 ± 0.30	-	**3.66 ± 0.60**	−16%	3.13 ± 0.33	-						
**D**		2.91 ± 0.38	-	3.27 ± 0.29	-	3.42 ± 0.31	-	3.13 ± 0.45	-						
**E**		2.95 ± 0.31	-	3.24 ± 0.18	-	3.36 ± 0.24	-	3.15 ± 0.45	-	-		-		-	
**F**		**2.78 ± 0.39**	12%	2.88 ± 0.61	-	**2.67 ± 0.57**	16%	**2.58 ± 0.44**	19%						
**G**		**2.68 ± 0.36**	15%	**2.62 ± 0.46**	17%	**2.73 ± 0.62**	21%	**2.38 ± 0.54**	25%						
**H**		**2.60 ± 0.34**	18%	**2.48 ± 0.58**	22%	**2.42 ± 0.49**	24%	**2.41 ± 0.34**	24%						
**I**		**2.54 ± 0.37**	20%	**2.51 ± 0.36**	21%	**2.36 ± 0.38**	26%	**2.40 ± 0.31**	22%						
**A**	50% methanol	2.98 ± 0.57	-	3.18 ± 0.43	-	3.52 ± 0.54	-	3.31 ± 0.77	-						
**B**		3.04 ± 0.52	-	3.37 ± 0.38	-	3.49 ± 0.42	-	3.28 ± 0.46	-						
**C**		3.10 ± 0.74	-	3.36 ± 0.24	-	3.66 ± 0.79	-	3.13 ± 0.80	-						
**D**		3.07 ± 0.46	-	3.24 ± 0.63	-	3.37 ± 0.57	-	2.99 ± 0.73	-						
**E**		3.02 ± 0.65	-	3.03 ± 0.58	-	3.24 ± 0.31	-	3.24 ± 0.44	-	-		-		-	
**F**		**2.37 ± 0.20**	25%	**2.46 ± 0.29**	22%	2.57 ± 0.41	19%	2.85 ± 0.50	-						
**G**		**2.54 ± 0.29**	20%	**2.57 ± 0.50**	19%	2.73 ± 0.47	14%	2.89 ± 0.46	-						
**H**		**2.41 ± 0.37**	24%	**2.82 ± 0.40**	11%	2.88 ± 0.49	-	2.98 ± 0.40							
**I**		2.87 ± 0.55	-	2.77 ± 0.78	-	2.88 ± 0.71	-	**2.49 ± 0.44**	22%						
**A**	water							**2.69 ± 0.55**	15%	2.75 ± 0.69	-	**2.70 ± 0.42**	14%	**2.49 ± 0.51**	21%
**B**								**2.58 ± 0.62**	18%	**2.68 ± 0.60**	15%	**2.69 ± 0.38**	14%	**2.38 ± 0.26**	24%
**C**								**2.66 ± 0.66**	15%	2.89 ± 0.56	-	**2.79 ± 0.38**	11%	**2.55 ± 0.28**	19%
**D**								**2.60 ± 0.56**	18%	3.17 ± 0.65	-	2.85 ± 0.45	-	**2.34 ± 0.27**	26%
**E**				-		-		**2.50 ± 0.53**	21%	3.04 ± 0.59	-	2.91 ± 0.31	-	**2.59 ± 0.21**	18%
**F**								**2.50 ± 0.51**	21%	**2.33 ± 0.52**	27%	**2.46 ± 0.41**	22%	**2.09 ± 0.29**	34%
**G**								**2.48 ± 0.43**	22%	**2.43 ± 0.42**	23%	**2.43 ± 0.28**	23%	**2.42 ± 0.34**	24%
**H**								**2.53 ± 0.36**	20%	**2.13 ± 0.27**	33%	**2.03 ± 0.32**	36%	**2.02 ± 0.23**	36%
**I**								**2.34 ± 0.25**	26%	**2.25 ± 0.29**	29%	**2.14 ± 0.33**	32%	**2.16 ± 0.39**	32%

Each value is expressed as mean ± standard deviation from at least three independent biological evaluations. Bold values denote statistical significance at the *p* < 0.05 level. A dash “-” denotes that the concentration has not been applied. The extracts from the following treatments were investigated: (**A**) control roots collected after 10 days of culture; (**B**) control roots collected after 20 days of culture; (**C**) control roots collected after 30 days of culture; (**D**) control roots collected after 40 days of culture; (**E**) control roots collected after 50 days of culture; (**F**) hairy roots elicited for 3 days with 50 µM MeJA; (**G**) hairy roots elicited for 7 days with 100 µM MeJA; (**H**) hairy roots elicited for 1 day with 100 µM MeJA 100 and then transferred to control medium for 30 day cultivation; (**I**) hairy roots elicited for 7 days with 100 µM MeJA 100 and then transferred to control medium for 30-day cultivation.

**Table 5 molecules-27-00186-t005:** Evaluation of antigenotoxic activity based on IR values for the control—genotoxin 2-aminoanthracene (2AA 10 µg/mL)—and the mixture of 2AA with extracts dissolved in methanol, 50% methanol, or deionized water.

Tested Extract						Concentration of Tested Extracts [µg/mL] IR Values (Mean ± SD)//%Antigenotoxicity									
	0	12.5		25		50		100		200		400		800	
**Control**	**3.25 ± 0.55**														
**A**	methanol	3.05 ± 0.93	-	2.77 ± 1.03	-	2.91 ± 1.13	-	2.78 ± 0.83	-						
**B**		2.92 ± 0.43	-	2.62 ± 0.81	19%	2.91 ± 0.93	-	3.00 ± 0.67	-						
**C**		2.98 ± 0.73	-	2.86 ± 1.03	-	3.10 ± 0.95	-	2.91 ± 0.50	-						
**D**		3.94 ± 0.83	−21%	2.72 ± 0.78	-	2.98 ± 0.60	-	**2.32 ± 0.30**	28%						
**E**		3.50 ± 0.41	-	3.80 ± 0.88	-	3.55 ± 1.04	-	3.16 ± 0.80	-	-		-		-	
**F**		2.55 ± 0.40	22%	2.15 ± 0.34	34%	**2.11 ± 0.49**	35%	**2.00 ± 0.26**	38%						
**G**		3.02 ± 0.71	-	2.91 ± 0.71	-	2.84 ± 0.92	-	**2.53 ± 0.97**	22%						
**H**		3.07 ± 0.60	-	3.65 ± 0.93	-	3.22 ± 0.61	-	2.87 ± 1.06	-						
**I**		3.06 ± 0.78	-	3.87 ± 1.31	-	3.01 ± 0.61	-	2.87 ± 1.13	-						
**A**	50% methanol	3.10 ± 0.86	-	**2.68 ± 0.58**	18%	**2.57 ± 0.92**	21%	3.11 ± 0.64	-						
**B**		3.18 ± 0.93	-	3.05 ± 1.01	-	2.92 ± 0.38	-	3.25 ± 0.38	-						
**C**		3.02 ± 0.48	-	3.32 ± 0.79	-	3.21 ± 0.60	-	3.06 ± 0.59	-						
**D**		2.99 ± 0.51	-	**2.39 ± 0.69**	26%	2.77 ± 0.76	-	**2.67 ± 0.75**	18%						
**E**		3.04 ± 0.34	15%	2.99 ± 1.14	-	2.98 ± 1.18	-	2.80 ± 0.95	-	-		-		-	
**F**		2.75 ± 0.45	17%	2.98 ± 0.66	-	3.17 ± 0.77	-	2.85 ± 0.85	-						
**G**		**2.71 ± 0.34**	-	3.95 ± 1.39	-	3.35 ± 0.98	-	2.89 ± 0.94	-						
**H**		2.97 ± 0.82	30%	3.06 ± 0.59	-	3.39 ± 0.71	-	2.89 ± 0.94	-						
**I**		**2.28 ± 0.27**	-	**2.29 ± 0.28**	30%	**2.28 ± 0.27**	30%	**2.13 ± 0.32**	34%						
**A**	water							3.72 ± 0.31	-	3.59 ± 0.73	-	2.89 ± 0.79	-	2.86 ± 0.66	-
**B**								3.26 ± 0.82	-	**2.50 ± 0.24**	23%	**2.48 ± 0.71**	24%	**2.44 ± 0.50**	25%
**C**								3.96 ± 1.43	-	3.24 ± 1.09	-	**2.40 ± 0.80**	26%	**2.54 ± 0.53**	22%
**D**								3.84 ± 1.21	-	2.84 ± 1.10	-	**2.38 ± 0.75**	27%	**2.58 ± 0.53**	20%
**E**				-		-		3.55 ± 0.86	-	2.95 ± 0.95	-	**2.55 ± 0.79**	21%	**2.65 ± 0.67**	18%
**F**								**2.34 ± 0.63**	28%	**2.04 ± 0.54**	37%	**1.97 ± 0.26**	39%	**1.55 ± 0.38**	52%
**G**								**2.51 ± 0.75**	23%	**2.31 ± 0.45**	29%	**2.51 ± 0.86**	23%	**1.85 ± 0.52**	43%
**H**								**2.48 ± 0.48**	24%	**2.44 ± 0.60**	25%	**2.35 ± 0.58**	28%	**2.01 ± 0.81**	38%
**I**								**2.70 ± 0.61**	17%	**2.47 ± 0.62**	24%	**2.46 ± 0.64**	24%	**1.65 ± 0.37**	49%

Each value is expressed as mean ± standard deviation from at least three independent biological evaluations. Bold values denote statistical significance at the *p* < 0.05 level. A dash “-” denotes that the concentration has not been applied. The extracts resulted from the following treatments were investigated: (**A**) control roots collected after 10 days of culture; (**B**) control roots collected after 20 days of culture; (**C**) control roots collected after 30 days of culture; (**D**) control roots collected after 40 days of culture; (**E**) control roots collected after 50 days of culture; (**F**) hairy roots elicited for 3 days with 50 µM MeJA; (**G**) hairy roots elicited for 7 days with 100 µM MeJA; (**H**) hairy roots elicited for 1 day with 100 µM MeJA 100 and then transferred to control medium for 30 day cultivation; (**I**) hairy roots elicited for 7 days with 100 µM MeJA 100 and then transferred to control medium for 30-day cultivation.

**Table 6 molecules-27-00186-t006:** Sequences of primers used in the PCR reaction [36].

Gene	Forward Primer	Reverse Started
*rol*B	5′-GCTCTTGCAGTGCTAGATTT-3′	5′-GAAGGTGCAAGCTACCTCTC-3′
*rol*C	5′-CTCCTGACATCAAACTCGTC-3′	5′-TGCTTCGAGTTATGGGTACA-3′
*vir*G	5′-ACTGAATATCAGGCAACGCC-3′	5′-GCGTCAAAGAAATAGCCAGC-3′

## Data Availability

All relevant data for the preparation of this manuscript are given in the text. Raw data used for preparation of this manuscript are available on request from the authors.

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
