# Peer review of "Polyscias filicifolia (Araliaceae) Hairy Roots with Antigenotoxic and Anti-Photogenotoxic Activity"

_molecules, 2021, doi:10.3390/molecules27010186_

Round 1

Reviewer 1 Report

The paper focuses on the description of composition, anti-genotoxic and anti-photogenotoxic activities of hairy roots of Polyscias filicifolia enhanced by methyl jasmonate elicitation.

General comment: The study is interesting but incomplete and requires additional experiments.

The biological activities are evaluated for all the extracts but methanol extract is the only one to be characterized. Please achieve phytochemical analyzes HPLC-PDA-ESI-MS on methanol 50% and water extracts.

Several compounds are identified in phytochemical analyzes. Please achieve pharmacological evaluation of compounds as chlorogenic acid, 3,5-dicaffeoylquinic acid or 5-O-feruoylquininic acid.

Specific comments:

The term specific metabolite is more suitable than secondary metabolite.

Italicize the name of plants and microorganisms

- Line 53: please add a reference to traditional use.

- Table 3: index the number in the formulas. Prefer the term “Condition” to “Extract type”.  “Extract type” should be used for the type of solvent. Widen the "Compound" column and decrease the "Peaks N °", "Tr", “[M-H]- “columns.

- Line 267: reformulate « The extracts showed the highest activity when dissolved in deionised water ». There is a misinterpretation between extracts obtained by one type of solvent and the solvent used for the test. This error appears in several parts of the text.

- Table 5 and 6 are the same!

- Tables 4, 5 and 6 can be combined in one table

- Tables 7,8 and 9 together can be combined in one table

Author Response

Reviewer #1
We would like to thank the Reviewer for providing such constructive criticism. The manuscript has 
been revised accordingly, and we feel the latest version has benefitted considerably. All suggestions 
have been incorporated in the revised manuscript, and these have been marked.
1) The biological activities are evaluated for all the extracts but methanol extract is the only one 
to be characterized. Please achieve phytochemical analyzes HPLC-PDA-ESI-MS on methanol 
50% and water extracts.
Thank you for this valuable remark. In fact it was not clear what type of extract was 
investigated. Now it was written more detailed in Results and Material and Methods sections. 
The only one type of extract, that is methanolic (100 %) was subjected for experiments 
described in current study. For biological activity investigations this extract was dissolved to 
obtained 50% methanolic solution or was dissolved in water. Also biological activity of 100% 
methanolic extract was investigated.
That is why the HPLC-PDA-ESI-MS of 100 % methanolic extracts was carried out.
2) Several compounds are identified in phytochemical analyzes. Please achieve pharmacological 
evaluation of compounds as chlorogenic acid, 3,5-dicaffeoylquinic acid or 5-O-feruoylquininic 
acid.
Thank you for this comment and suggestion. This is the plan for our further investigation as 
present study is the first report on Polyscias filicifolia hairy roots, their chemical profile and 
biological activity.
3) The term specific metabolite is more suitable than secondary metabolite.
It was done. Only in original titles of articles it was left: secondary metabolites.
4) Italicize the name of plants and microorganisms.
It was done. Only in original titles of articles it was left: Araliaceae.
5) Line 53: please add a reference to traditional use.
It was done in line 57, reference number 8.
6) Table 3: index the number in the formulas. Prefer the term “Condition” to “Extract type”.
“Extract type” should be used for the type of solvent. Widen the "Compound" column and 
decrease the "Peaks N °", "Tr", “[M-H]-
“columns.
It was done.
7) Line 267: reformulate « The extracts showed the highest activity when dissolved in deionised 
water ». There is a misinterpretation between extracts obtained by one type of solvent and the 
solvent used for the test. This error appears in several parts of the text.
The only one type of extract prepared from hairy roots cultivated in various culture variants 
were subjected to antigenotoxic and anti-photogenotoxic activity investigations. The dry 
residue of the methanolic extract (100%), prior to experiments, was dissolved in deionized 
water or in 50% methanol and 100% methanol. These three types of solvents were used to 
test whether they would affect the activity due to different degrees of solubility of the 
compounds. Furthermore, methanol is toxic to the bacteria used in the umu test. The use of 
water as a solvent allowed higher concentrations of extracts to be tested.
8) Table 5 and 6 are the same!
Tables 4, 5 and 6 can be combined in one table
Tables 7,8 and 9 together can be combined in one table
Table 4, corrected 5 and 6 were combined in one Table 4 .
Table 7, 8 and 9 were combined in one Table 5 .

Reviewer 2 Report

Line 44-46 : Please provide reference.

Line 52-54: Please provide reference.

Line 76: Agrobacterium rhizogenes should be in italic. Check throughout.

Line 77 : Agrobacterium rhizogenes LBA 9402 strain 77.

Line 374: Which type of bioactive compounds?

Line 462: Please add the composition of hormone-free MS medium.

Line 455: Please justify the use acetosyringone in the experiment.

Line 652- 654:Justify the use of the two statistical tests and what is the difference between the performed tests ?

Author Response

Reviewer #2
We would like to thank the Reviewer for providing such constructive criticism. The manuscript has 
been revised accordingly, and we feel the latest version has benefitted considerably. All suggestions 
have been incorporated in the revised manuscript, and these have been marked.
1) Line 44-46 : Please provide reference.
The following reference number 6 is given line 49: Ramirez-Estrada, K.; Vidal-Limon, H.; 
Hidalgo, D.; Moyano, E.; Goleniowski, M.; Cusidó, R. M. Elicitation, an effective strategy for the 
biotechnological production of bioactive high-added value compounds in plant cell factories. 
Molecules 2016, 21, doi:10.3390/molecules21020182.
2) Line 52-54: Please provide reference
It was done in line 57, reference number 8.
3) Line 76: Agrobacterium rhizogenes should be in italic. Check throughout.
It was done.
4) Line 77 : Agrobacterium rhizogenes LBA 9402 strain 77.
It was done.
5) Line 374: Which type of bioactive compounds?
The following compounds and their mixtures with Polyscias filicifolia: photogenotoxin 
chlorpromazine (CPZ 5 µg/mL) —and the mixture of CPZ with extracts (100 µg/mL) dissolved in 
methanol,50% methanol or deionized water.
6) Line 462: Please add the composition of hormone-free MS medium.
MS medium, that is Murashige et Skoog (1962), is widely used and popular medium for 
cultivation of plant cells and its composition is available in publication cited as well as in various 
catalogues that offer ready-to-use medium salts. Hormone-free means that without any plant 
growth regulators eg. auxins or cytokinins.
7) Line 455: Please justify the use acetosyringone in the experiment.
This was added.
8) Line 652- 654:Justify the use of the two statistical tests and what is the difference between the 
performed tests ?
In our study data with normal distribution was analyzed using Student’s t-test, while this 
without normal distribution were analyzed with the nonparametric U Mann– Whitney method.

Reviewer 3 Report

This research is interesting and provides significant areas using Polyscias filici hairy root cultures leaf (Araliaceae) with Agrobacterium rhizogenes transformation ATCC 15834. It presents the results of a study where 15 root lines could be obtained by transformation, with this strategy resulting in chlorogenic acid production and other different caffeic and ferulic acid derivatives without cytotoxicity. In addition these compounds demonstrated significant antigenotoxic and anti-photogenotoxic properties.
The manuscript is well written and presented. I have no reservations about it, but an extended conclusion would be more appropriate for a better understanding.

Author Response

Reviewer #3
I have no reservations about it, but an extended conclusion would be more appropriate for a better 
understanding.
We would like to thank the Reviewer for this remark. We did our best to improve the Conclusion section
and these have been marked. We feel the latest version has benefitted considerably.

Round 2

Reviewer 1 Report

The names of families like Araliaceae are not in italics, only the names of genera and species.